# A Rare Evolution to Pneumopericardium in Patient with COVID-19 Pneumonia Treated with High Flow Nasal Cannula

**DOI:** 10.3390/medicina57101122

**Published:** 2021-10-18

**Authors:** Giorgio Emanuele Polistina, Maurizia Lanza, Camilla Di Somma, Anna Annunziata, Giuseppe Fiorentino

**Affiliations:** Sub-Intensive Care Unit and Respiratory Physiopathology Department, Cotugno—Monaldi Hospital, 80121 Naples, Italy; maurizia.lanza85@gmail.com (M.L.); camilladisomma@gmail.com (C.D.S.); anna.annunziata@gmail.com (A.A.); giuseppefiorentino1@gmail.com (G.F.)

**Keywords:** pneumopericardium, coronavirus, Covid-19, HFNC, pneumonia

## Abstract

Infection with severe acute respiratory syndrome coronavirus 2 causes coronavirus disease 2019 (COVID-19) which was revealed an official pandemic by the World Health Organization on 11 March 2020. The current pandemic, the third of this decade, is the worst in terms of suffering and deaths related. COVID-19 represents an unprecedented challenge for medical communities and patients around the world. High-resolution computed tomography of the chest (HRCT) is a fundamental tool in both management and diagnosis of the disease. Imaging plays an essential role in the diagnosis of all the manifestations of the disease and its complications and the correct use and interpretation of imaging tests are essential. Pneumomediastinum has been reported rarely in COVID-19 patients. We were one of the first groups to share our experiences in uncommon parenchymal complications of COVID-19 with spontaneous pneumothorax and pneumomediastinum, but also with new-onset bronchiectasis and cysts. A finding of pneumopericardium is also unusual. We hereby report a rare case of spontaneous pneumopericardium in a patient with COVID-19 pneumonia treated only with a high-flow nasal cannula (HFNC).

## 1. Introduction

Severe acute respiratory syndrome coronavirus 2 (SARS-CoV-2) is characterized by severe bilateral pneumonia with complication leading to hypoxic respiratory failure, acute respiratory distress syndrome (ARDS), cytokine storm, disseminated intravascular coagulation (DIC) and from gastrointestinal, neurological and cutaneous syndromes [1]. Imaging plays an essential role in diagnosing all manifestations of the disease and its related complications, and the correct use and interpretation of imaging tests is crucial. Typical radiological signs of pneumonia caused by COVID-19 are the ‘ground-glass opacities and consolidations but are also frequently encountered other atypical radiographic features as: isolated lobar or segmental consolidation without ground glass opacity (GGO), small discrete nodules (centrilobular, “shaft in bud”), pulmonary cavitation and thickening of the smooth interlobular septum with pleural effusion [2,3].

To date, pneumomediastinum has been considered a probable rare complication of COVID-19 pneumonia [4]. Sporadic cases of pneumopericardium associated with COVID-19 pneumonia have been reported in the Literature [5], none of which associated with High Flow Nasal Cannula oxygen support.

Pneumopericardium is generally a rare condition, which may be associated with pneumomediastinum. The main causes may be associated with positive pressure ventilation, chest surgery, trauma, infectious pericarditis, or fistula between the pericardium and an adjacent air-containing organ [6].

The mechanism of development of this complication probably in the COVID-19 pneumonia is poorly understood and our report seeks to explore it. 

The implications of these results in terms of indicators of disease severity and impact on clinical outcome open up an important space for further discussions.

## 2. Case Report

A 70-year-old Hispanic male with a previous history of hypertension, pericarditis completely resolved about seven years ago and recent onset of Parkinson’s disease presented to the emergency department with one-week history of fever, chills, and worsening shortness of breath. Lifelong nonsmoker, and no history of illicit substance abuse and body mass index (BMI) of 23 kg/m^2^. No signs of cardiological distress were detected on admission.

At presentation, he resulted positive for the novel SARS-CoV-2 on swab test in real time polymerase chain reaction assay. His baseline vital signs were temperature of 38 degrees Celsius, heart rate of 110 beats per minute (BPM), respiratory rate 32/min with an oxygen saturation of 82% on room air. Blood pressure was 130/80 mm of mercury (mmHg). Physical examination revealed tachypnea with clear lungs on auscultation and no visible signs of respiratory distress. 

Heart auscultation demonstrated no identifiable rubs, murmur, or bruits on cardiovascular examination.

Laboratory examination revealed mild leukocytosis (12.56 × 10^3^/µL) with lymphopenia (0.3 × 10^3^/µL), elevated C-reactive protein (CRP) and Dimer D (7.2 mg/dL, 3190 IU/L, respectively). Levels of interleukin-6 (IL-6) were imperceptibly higher 42.7 pg/mL, no other anomalies were found. Bacterial cultures and the PCR for other respiratory viruses panel were negative. 

The high-resolution computed tomography scan (HRTC) made on admission (Figure 1) revealed large consolidations and bilateral ground glass opacities (GGO) areas such as progressive Covid pneumonia scored with 11/20 on Total severity score (TSS) [7].

Due to the significant deterioration of the patient’s respiratory function he was transferred to a negative pressure isolation room and treated with high flow nasal cannula (HFNC) with FiO_2_ concentration of 60% and 60 l/min to reach Spo2 of 97%. The patient responded well to treatment and showed on arterial blood gas (ABG) PaO2/FiO2 ratio of 150. Stat electrocardiogram (EKG) on admission showed sinus rhythm at HR 60 bpm. Sign of slight left axial deviation. Atrio-ventricular conduction at the high limits with PR interval 200 msec. Intraventricular conduction within the limits. (Figure 2A).

Patient was treated with intravenous corticosteroids and Remdesivir© for five days with partial improvement of clinical conditions. 

On day 12 computed tomography scan (CT) performed due to elevation of the dimer D at blood test revealed small layer of pneumomediastinum bubble with an increase in consolidation component without any clinician symptoms (Figure 3).

On day 22 from admission on a control HRCT was noted massive pneumopericardium in the absence of related symptoms (Figure 4). A new EKG was performed with absence of significant voltage reduction and showed sinus rhythm at HR 70 bpm. Atrioventricular conduction with PR 200 msec. Intraventricular conduction within the limits. (Figure 2B). 

A gradual reduction of pneumopericardium was observed on a control HRTC until a complete resolution of air leaks. The patient was not subjected to any surgical therapy without any cardiovascular complications with progressive decrease of oxygen supplementation and flow until discharge. 

## 3. Discussion

In March 2021, we observed the first case of evolution to pneumopericardium with pneumomediastinum (PM) in a patient with COVID-19 pneumonia treated only with high flow nasal cannula (HFNC). Different uncommon parenchymal complications of COVID-19 have been described over time with spontaneous pneumothorax and pneumomediastinum, but also with new-onset bronchiectasis and cysts not associated with mechanical ventilation [8].

Spontaneous pneumopericardium is more singular than pneumomediastinum. So far, it has not been possible to discover any case in the literature that clarifies PM’s temporal and radiological evolution to pneumopericardium in HFNC treatment in COVID-19 pneumonia. Sporadic cases of spontaneous pneumothorax, PM and pneumopericardium are described in the literature to establish the unusual complication [9,10].

Pneumopericardium is a rare syndrome characterized by the abnormal presence of air in the pericardial sac. In patients with pneumopericardium this leads to an increased risk of morbidity and mortality. DDA a clinical point of view, there are two forms of pneumopericardium: not stress and tension. The pneumopericardium tension is manifested by a decrease in cardiac output and consequent circulatory failure. The pneumopericardium is a rare syndrome that recognizes various etiologies. It may be associated with another form of air leak syndrome, such as pneumothorax, pneumomediastinum, pulmonary interstitial emphysema and subcutaneous emphysema. 

In non—tension pneumopericardium, the air inside the pericardial cavity is present but does not cause complications. 

In contrast, a pneumopericardium under tension during air pressure prevents cardiac contraction and, consequently, reduces cardiac output. All cardiac examinations are described signs and specific symptoms. Typically, the signs of tension pneumopericardium include paradoxical pulses, tachycardia, decreased cardiac output with increased central venous pressure, muffled heart sounds, and low-voltage ECG traces [11].

*Macklin* first suggested the potential mechanisms of pneumopericardium development. It develops when there is direct communication between the pericardium and the airways, pleuropericardial communication with a pneumothorax, or air movement from the lung parenchymal tissue conducting the pulmonary perivascular sheaths to the pericardium due to alveolar rupture (Macklin effect) [12]. Patients with tension pneumopericardium often have no other air leak syndromes and therefore, patients who manifested with cardiac tamponade more often had no other associated air leaks. Solitary pneumopericardium is most often tension because when linked to other air leak syndromes, the air is not confined to the closed space of the pericardial sheet; it can instead be redistributed to different adjacent compartments (e.g., mediastinum and pleura), preventing and minimizing the occurrence of a prolongation of local pericardial pressure.

The cytokine storm involved in the pathophysiology of COVID-19 could be responsible for the diffuse alveolar injury that makes the alveoli more prone to rupture. Alveolar rupture can be induced by severe cough in spontaneously breathing patients or by barotrauma in mechanically ventilated patients due to the frequent use of high positive-end expiratory pressure [11].

We hypothesize that this dysregulated inflammation could potentially sever the barrier between the pericardium and the mediastinum and if a pneumomediastinum is present, there would be communication for air development. 

The decision whether or not to treat pneumopericardium depends on whether or not cardiac tamponade is started.

The management of isolated asymptomatic pneumopericardium requires careful monitoring not accompanied by therapeutic interventions, if the presence of tension pneumopericardium is hypothesized, it is essential to proceed with pericardial aspiration. This manoeuvre is required to ensure that irreversible hemodynamic changes do not occur and can impair the pumping function of the heart muscle.

If the pneumopericardium is particularly extensive or recurrent in its presentation, inserting a drainage tube into the pericardial sac may be necessary.

Reassembling the techniques for managing this complication, we can say that:Tension-free pneumopericardium should be carefully monitoredTension-free pneumopericardium has a much higher survival rate than patients with cardiac tamponade.Factors such as recent surgery, a history of hematological disease, and isolated pneumopericardium should be recognized as warning signs and may favor the emergence of tension pneumopericardium and therefore the need for immediate intervention is recommended in these patients.

## 4. Conclusions

COVID 19 mortality is determined by the different complications that patients develop; nevertheless, some remain underreported. Clinicians and radiologists should be advised of the possibility of evolution of pneumothorax, PM and pneumopericardium as a severe complication of COVID-19 pneumonia that can occur even if treated only with HFNC and can affect patient management and clinical outcomes. The exact mechanism for which this disruption occurs in COVID-19 patients remains unclear. Clinicians should also consider these complications, that are not extremely rare, in relation to the choice of the best ventilatory support. 

We reported our conservative management, with gradual resorption of the air from the tissues. 

We need more data to evaluate possible causation of pneumomediastinum and pneumopericardium secondary to COVID-19.

## Figures and Tables

**Figure 1 medicina-57-01122-f001:**
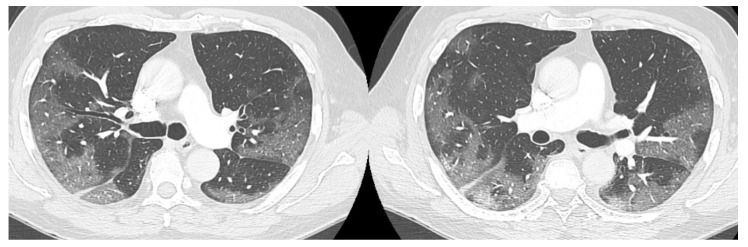
HRTC performed on admission with large consolidations and bilateral ground glass opacities (GGO) TSS 11/20.

**Figure 2 medicina-57-01122-f002:**
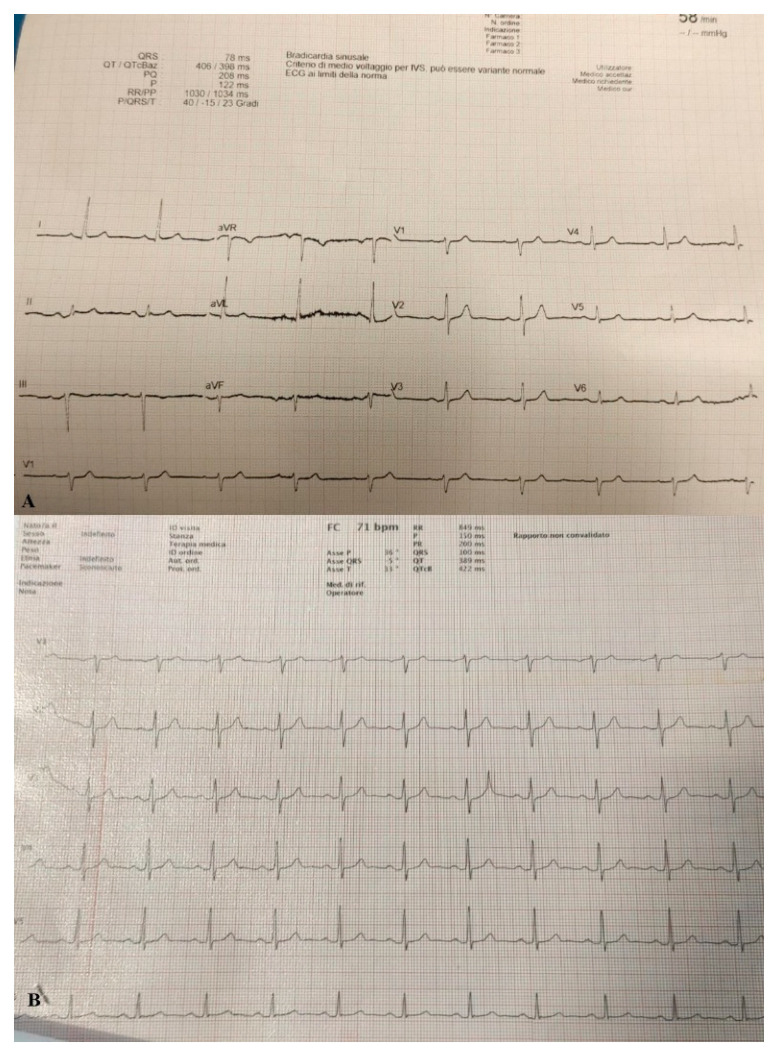
(**A**) EKG performed on admission. (**B**) EKG performed when pneumopericardium was discovered.

**Figure 3 medicina-57-01122-f003:**
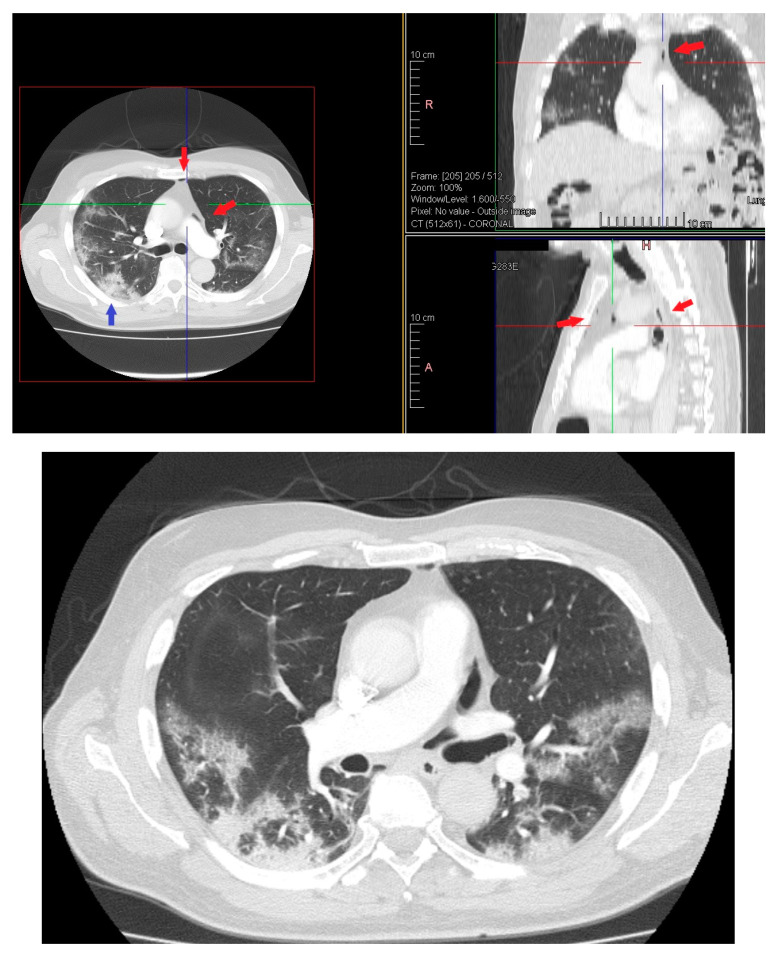
HRTC performed on day 12 showing pneumomediastinum bubble (red arrows) and increase in consolidation component (blue arrow). Detail of HRTC performed on day 12 with small layer of pneumomediastinum bubble.

**Figure 4 medicina-57-01122-f004:**
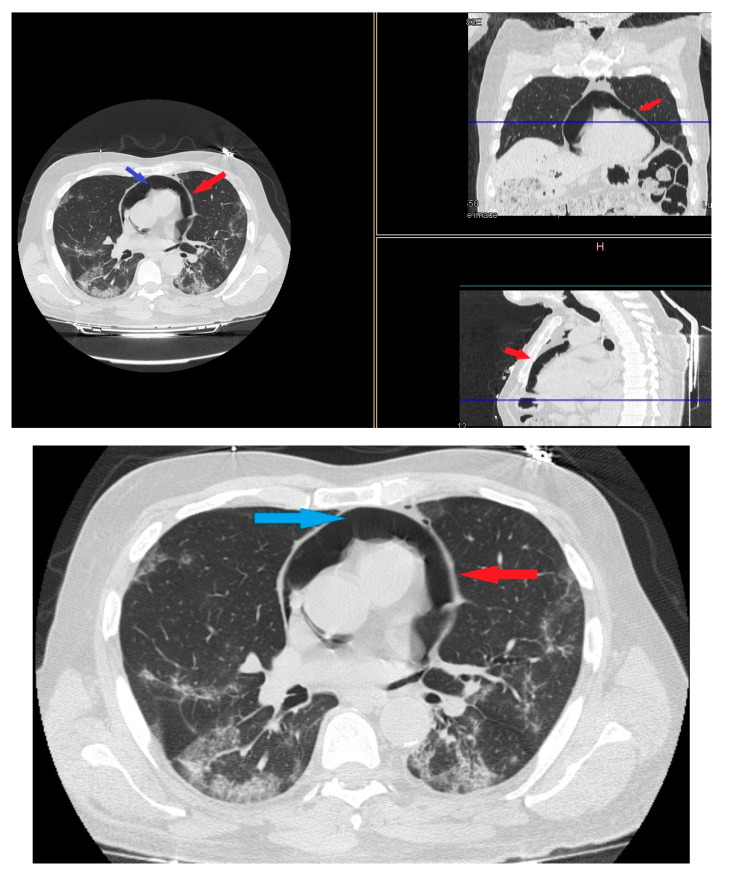
HRTC performed on day 22 from admission with evidence of massive pneumopericardium (red arrows) and severe presence of gas inside the pericardial sac (blue arrow). Detail of HRTC performed on day 22 show evidence of pneumopericardium (red arrow) and the severe presence of gas in the pericardium (blue arrow).

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
