# Peer review of "A Rare Evolution to Pneumopericardium in Patient with COVID-19 Pneumonia Treated with High Flow Nasal Cannula"

_medicina, 2021, doi:10.3390/medicina57101122_

Round 1
Reviewer 1 Report
Dear Authors
There are several cases and case series published on pneumomediastinum and some but less on pneumopericardium in COVID-19 e.g.:
Juárez-Lloclla JP, León-Jiménez F, Urquiaga-Calderón J, Temoche-Nizama H, Bryce-Alberti M, Portmann-Baracco A, Bryce-Moncloa A. Spontaneous Pneumopericardium and Pneumomediastinum in Twelve COVID-19 Patients. Arch Bronconeumol. 2021 Jan;57 Suppl 1:86-88. doi: 10.1016/j.arbres.2020.09.013. Epub 2020 Oct 9. PMID: 33158641; PMCID: PMC7546662.
Murillo Brito, Diego Alejandro; Villalva, Camila Elena Arranz; Simón, Ana Suárez; Guisado-Clavero, Marina (2020): COVID-19 Pneumonia Associated With Spontaneous Pneumomediastinum and Pneumopericardium. CTSNet, Inc. Media. https://doi.org/10.25373/ctsnet.12485831.v1
In you case you refer to previous pericarditis in a patient with Parkinson's disease. You did not comment on whether the patient was treated with cabergoline with has been describe as associated with constrictive pericarditis. https://heart.bmj.com/content/heartjnl/90/8/e47.full.pdf You also did not refer whether there was any residual effects on the pericard. It will be useful to get a focussed review of the imaging to check for any thickening etc. Noted that the air did not track all the way around the right side of the heart.
The image quality can be improved and my observations are in context of suboptimal image quality. You refer to isolated pneumopericardium but on the coronal view Figure 4 there may be air in the left side of the superior mediastinum. Is this not a pneumomediastinum with a secondary pneumopericardium. A review of the air in the sagittal view also suggested. There is a subtle hard border to the postro-medial pleural interface which was not obvious in figure 3. Obtain a review of this as there may be a minute bit of air tracking.
The ECG image quality is inadequate. Why is there a 12 lead ECG in Figure 2 A and multi lead rhythm strip in B?
There are more abnormalities in A than quoted (difficult to interpret due to quality of the image) but additional to the LAD a suggestion of PR depression in several leads and subtle ST changes. The lateral chest lead voltage also seems a bit low.
This may be an interesting (although not fully novel) case, from which colleagues can learn. The issues as above however will need to be addressed in detail.
This case description overall needs to tighten up in attention to detail.
Author Response
Dear Reviewers,
Thank you for reviewing the case report titled "A rare evolution to pneumopericardium in patient with COVID-19 pneumonia treated with high flow nasal cannula."
Below are the detailed responses to the reviewer's questions.
I have also updated the original article and highlighted the changes that the reviewer's requested.

Reviewer 2 Report
I have read your article with great interest. This is an important case report. Just one thing needs to be fixed. The figure (ECG) is unclear. You need to scan the ECG instead of taking phot.Author Response
Dear Reviewers,
Thank you for reviewing the case report titled "A rare evolution to pneumopericardium in patient with COVID-19 pneumonia treated with high flow nasal cannula."
Below are the detailed responses to the reviewer's questions.
I have also updated the original article and highlighted the changes that the reviewer's requested.
Unfortunately, the EKG images cannot be improved. Poor quality results from the inability to move contaminated material into non-covid areas.
The two EKGs were performed in two different Departments.
EKG A was performed at the emergency department on admission.
EKG B was performed in our department after evidence of pneumopericardium on HRTC.

Round 2
Reviewer 1 Report
"DISCUSSION
In March 2021, we observed the first case of evolution to pneumopericardium without pneumomediastinum (PM) in a patient with COVID-19 pneumonia treated only with high flow nasal cannula (HFNC). "
In this case the pneumopericardium was preceded by pneumomediastinum therefore the first sentence of the discussion is incorrect. The ongoing connection between the mediastinum and pericardium may have played a role in preventing a tension pneumopericardium as referred to later in the text.
Author Response
Dear Reviewers,
Thank you for reviewing the case report titled "A rare evolution to pneumopericardium in patient with COVID-19 pneumonia treated with high flow nasal cannula."
I have also updated the original article and highlighted the changes that the reviewer's requested.
